# Hypothetical acceptability of hospital-based post-mortem pediatric minimally invasive tissue sampling in Malawi: The role of complex social relationships

**Sarah Lawrence** [1]*, **Dave Namusanya**[2], **Andrew Hamuza**[2], **Cornelius Huwa**[3,4], **Dennis Chasweka**[3,4], **Maureen Kelley**[4,5], **Sassy Molyneux**[4,5], **Wieger Voskuijl**[3,4,6,7], **Donna M. Denno**[1,4,8], **Nicola Desmond**[2,9]

1 Department of Pediatrics, University of Washington, Seattle, Washington, United States of America, 2 Behaviour and Health Research Group, Malawi-Liverpool-Wellcome Trust, Blantyre, Malawi, 3 Department of Paediatrics and Child Health, College of Medicine, Blantyre, Malawi, 4 The Childhood Acute Illness & Nutrition (CHAIN) Network, C/o KEMRI Wellcome Trust Research Programme, Nairobi, Kenya, 5 Wellcome Centre for Ethics & Humanities and Ethox Centre, Nuffield Department of Population Health, University of Oxford, Oxford, United Kingdom, 6 Department of Global Health, Amsterdam Institute for Global Health and Development, Amsterdam University Medical Centres, Amsterdam, The Netherlands, 7 Amsterdam Centre for Global Child Health, Amsterdam University Medical Centres, Amsterdam, The Netherlands, 8 Department of Global Health, University of Washington, Seattle, Washington, United States of America, 9 Department of International Public Health, Liverpool School of Tropical Medicine, Liverpool, United Kingdom

☯ These authors contributed equally to this work.

* sarahl34@uw.edu

**Data Availability Statement:** All focus group discussion files are available from the Figshare database (accession numbers https://doi.org/10.

## Abstract

### Background

Child mortality rates remain unacceptably high in low-resource settings. Cause of death (CoD) is often unknown. Minimally invasive tissue sampling (MITS)–using biopsy needles to obtain post-mortem samples–for histopathological and microbiologic investigation is increasingly being promoted to improve child and adult CoD attribution. "MITS in Malawi" is a sub-study of the Childhood Acute Illness & Nutrition (CHAIN) Network, which aims to identify biological and socioeconomic mortality risk factors among young children hospitalized for acute illness or undernutrition. MITS in Malawi employs standard MITS and a novel post-mortem endoscopic intestinal sampling approach to better understand CoD among children with acute illness and/or malnutrition who die during hospitalization.

### Aim

To understand factors that may impact MITS acceptability and inform introduction of the procedure to ascertain CoD among children with acute illness or malnutrition who die during hospitalization in Malawi.

### Methods

We conducted eight focus group discussions with key hospital staff and community members (religious leaders and parents of children under 5) to explore attitudes towards MITS

6084/m9.figshare.13335365.v1, https://doi.org/10.6084/m9.figshare.13335425.v1, https://doi.org/10.6084/m9.figshare.13337567.v1, https://doi.org/10.6084/m9.figshare.13338467.v1, https://doi.org/10.6084/m9.figshare.13337615.v1, https://doi.org/10.6084/m9.figshare.13337612.v1, https://doi.org/10.6084/m9.figshare.13338632.v1, https://doi.org/10.6084/m9.figshare.13338635.v1).

**Funding:** This study was funded by the Bill & Melinda Gates Foundation (https://www.gatesfoundation.org/) as a sub-study, led by DD and WV, of the CHAIN Network (grant number OPP1131320). Additional funding for MK and SM provided by a Wellcome Trust and MRC Newton Fund Collaborative Award (grant number 200344). The funders had no role in study design, data collection and analysis, decision to publish, or preparation of the manuscript.

**Competing interests:** The authors have declared that no competing interests exist.

and inform consent processes prior to commencing the MITS in Malawi study. We used thematic content analysis drawing on a conceptual framework developed from emergent themes and MITS acceptability literature.

## Results

Feelings of power over decision-making within the hospital and household, trust in health systems, and open and respectful health worker communication with parents were important dimensions of MITS acceptability. Other facilitating factors included the potential for MITS to add CoD information to aid sense-making of death and contribute to medical knowledge and new interventions. Potential barriers to acceptability included fears of organ and blood harvesting, disfigurement to the body, and disruption to transportation and burial plans.

## Conclusion

Social relationships and power dynamics within healthcare systems and households are a critical component of MITS acceptability, especially given the sensitivity of death and autopsy.

## Introduction

Child deaths in low- and middle-income countries have declined considerably over the past two decades. Despite these gains, accelerated progress and focused attention on vulnerable populations, including young children and those with poor nutritional status, is needed to achieve the Sustainable Development Goal (SDG) child health target (<25 deaths among children under age five years per 1000 live births) [1]. Malawi has one of the highest child mortality rates globally– 49.6 deaths per 1000 live births [2]. Enhanced understanding of risk factors for mortality and cause of death (CoD) determination will be important for improved development and application of interventions to reduce preventable pediatric fatality.

The Childhood Acute Illness & Nutrition (CHAIN) Network aims to identify biological and socioeconomic mortality risk factors among young children (2–23 months) hospitalized for acute illness or undernutrition across nine sites in six countries [3]. Children with moderate and severe wasting or kwashiorkor are purposefully oversampled in order to enhance the capacity to examine this vulnerability. Minimally invasive tissue sampling (MITS) is a CHAIN substudy at the Blantyre, Malawi site that aims to improve CoD understanding to tailor future intervention development. In MITS, small needles are used to sample organs and body fluids on which histopathological and microbiologic investigations are performed [4]. MITS in Malawi will also employ novel post-mortem endoscopic gastrointestinal sampling to assess the contribution of gastrointestinal pathology to mortality among acutely ill and undernourished children [5].

Multiple pathophysiologic processes underlie childhood illness and particularly malnutrition; identification of patients at risk of deterioration and death is difficult, and CoD determination is particularly challenging. Complete diagnostic autopsy (CDA) has long been considered the gold standard to establish CoD, however, cultural, social, and health system restrictions often render full autopsy impossible [6,7]. MITS is a valid alternative to CDA to establish CoD, including in pediatric deaths. MITS also addresses barriers to full autopsy, such

as concerns of bodily disfigurement [8–14]. While MITS has proven a valid alternative to CDA with the potential to address barriers of the procedure, acceptability of MITS remains context-specific and needs to be explored prior to implementation.

Acceptability in healthcare research and delivery is often poorly defined and ambiguously measured [15–19]. Consent or refusal to participate in research and patient satisfaction post-care seeking are often used as proxy measures of acceptability [15,18]. However, these approaches do not inform the degree of acceptability, factors that inform acceptability, or changes in acceptability over time [18]. Furthermore, consent and reported satisfaction reflect power relations and expectations more than acceptance.

Acceptability as a relative concept comparing the favourability of one intervention with another [19–21] has been utilized to measure willingness to consent for MITS in contrast to full autopsy through questionnaires [22,23]. While valuable, this approach limits insight into acceptability of this novel procedure since it draws largely on quantitative survey data. Qualitative studies have also been conducted to ascertain MITS acceptability in low-resource settings, including in Sub-Saharan Africa [6,24–29]. Some have primarily focused on community-based deaths, where families often have no prior diagnosis to inform CoD and are left seeking answers [6,24]. These studies suggest a range of potential benefits as well as context-specific sensitivities and barriers to MITS. Given context-specific findings in past literature and known beliefs and practices surrounding child-raising and death in Malawi, it would have been risky to assume MITS acceptability in the Malawian context prior to implementation [30–32].

The MITS in Malawi study will focus on deaths within the context of an ongoing hospital-based cohort study, CHAIN, where parents have been provided with pre-death clinical diagnoses. In this study we defined acceptability as both an ethically and socially embedded construct that includes absence of harm, positive effect, personal benefit and anticipated perceptions of usefulness and framed this within the social, cultural, and religious context of Southern Malawi. This qualitative study aimed to assess the hypothetical acceptability of MITS and factors informing acceptability in the Malawian context in order to inform the design of optimal ethical strategies for consent-seeking and other study processes prior to commencement of the MITS in Malawi study.

## Methods

### Study site

The CHAIN site in Malawi recruited patients from Queen Elizabeth Central Hospital (QECH), the central referral hospital in Blantyre, Southern Malawi, largely serving a poor patient population with services mostly provided free of charge [3]. Eligibility for CHAIN and the MITS in Malawi study included residence within a 50-kilometre radius of QECH, encompassing both urban and rural communities, including high density urban informal settlements and surrounding agricultural areas linked by high rates of mobility and income-driven migration. Southern Malawi is increasingly ethnically diverse but dominated by matrilocal ethnic groups such as the Chewa who are largely Christian.

### Study design and sampling

We utilized focus group discussions (FGD) to understand MITS acceptability and influencing factors. We reviewed the MITS consent and acceptability literature and interviewed researchers experienced in conducting pediatric autopsy and MITS in Mozambique [9,33] and pediatric autopsy and minimally invasive brain sampling in Malawi [34] to develop FGD topic guides in English, which were translated to Chichewa (the local language).

All FGD participants were purposively sampled to capture a wide variety of perspectives regarding MITS acceptability across strata. FGD participants were selected through local leaders, all of whom have pre-existing relationships with the Communication of Science Department at the Malawi-Liverpool-Wellcome Trust, the home institution of the social scientist study collaborators (DN, AH, ND). FGDs with health care workers (HCWs), nonclinical hospital support staff, and CHAIN research team members were held at QECH and focused on their: (1) perceptions of autopsy and MITS; (2) concerns regarding the use of MITS on children, including sampling of specific body parts; and (3) parental/communal concerns related to MITS and how to address them. The CHAIN team FGD also addressed potential implications of MITS on the CHAIN study. We included physicians (generalists and trainees), clinical officers, and nurses (including senior nurses) in the HCW FGD; guards, cleaners, patient attendants in the hospital support staff FGD; and the CHAIN study coordinator, study team clinicians, and fieldworkers in the CHAIN FGD. CHAIN and CHAIN Malawi site principal investigators were excluded from the CHAIN FGD.

FGDs with religious leaders from prevalent religions in the Blantyre area (Adventist, Catholic, Pentecostal, and Presbyterian Christian denominations and Islam) were conducted at health centres and focused on understanding: (1) religious norms and practices around death; (2) perceptions of autopsy; (3) attitudes towards MITS, especially for children and the sampling of specific body parts; and (4) concerns their respective congregations might have towards MITS, and how these concerns might be addressed. Religious leaders affiliated with apostolic churches that prohibit medical care seeking were excluded.

FGDs with mothers and fathers were held separately in health centres in two phases. First, parents who are health promotion volunteers (supported by HCWs to promote health practices and interventions in their communities) were selected because they are knowledgeable about acceptable practices in their communities and a bridge between the health system and the community. Two subsequent FGDs included parents without this tie to the health system to broaden our understanding of community perceptions towards MITS. All parent FGDs focused on: (1) cultural norms and practices around death; (2) perceptions of autopsy; (3) previous experiences with blood and other bodily fluid sampling in hospitals; (4) attitudes towards MITS for children, including sampling of specific body parts; (5) how to optimize the informed consent approach; and (6) potential community concerns about MITS, and how they might be addressed. Parents with no higher than secondary school education were recruited to reflect the population who seek healthcare at QECH. Parents with a hospitalized family member at the time of the FGD were excluded for ethical reasons.

## Data collection and analysis

Eight FGDs were conducted by a post-master's Malawian social scientist (DN) with six to 11 participants, lasting 79–92 minutes, between February and April 2018 (Round 1) and December 2018 (Round 2) (Table 1).

Semi-structured discussion guides (see S1–S6 Appendices) were utilized during FGDs to address key questions of interest. FGDs were primarily conducted in Chichewa, except for those with HCWs and CHAIN team members, which were mostly held in English, the official language in Malawian healthcare settings. FGDs were recorded, transcribed and translated verbatim. Translated transcripts were reviewed by the interviewer (DN) and a research assistant (AH) for accuracy and uploaded to NVivo 11 Plus (QSR International Pty Ltd). Our initial codebook was developed from an open coding process derived from the data. Additions were made as the analysis proceeded. All transcripts were independently coded by two of three coders (AH, DN, SL) and later merged. Thematic content analysis was used to iteratively develop

**Table 1. Overview of FGD participants.**

|  | Category | Number of participants | Date of FGD |
|---|---|---|---|
| **Hospital staff** | HCWs | 6 | February 2018 |
|  | Hospital support staff | 8 | March 2018 |
|  | CHAIN research team | 6 | March 2018 |
|  | **Total** | 20 |  |
| **Community Members** | Mothers Round 1 | 8 | March 2018 |
|  | Mothers Round 2 | 11 | December 2018 |
|  | Fathers Round 1 | 9 | March 2018 |
|  | Fathers Round 2 | 9 | December 2018 |
|  | Religious leaders | 8 | April 2018 |
|  | **Total** | 45 |  |

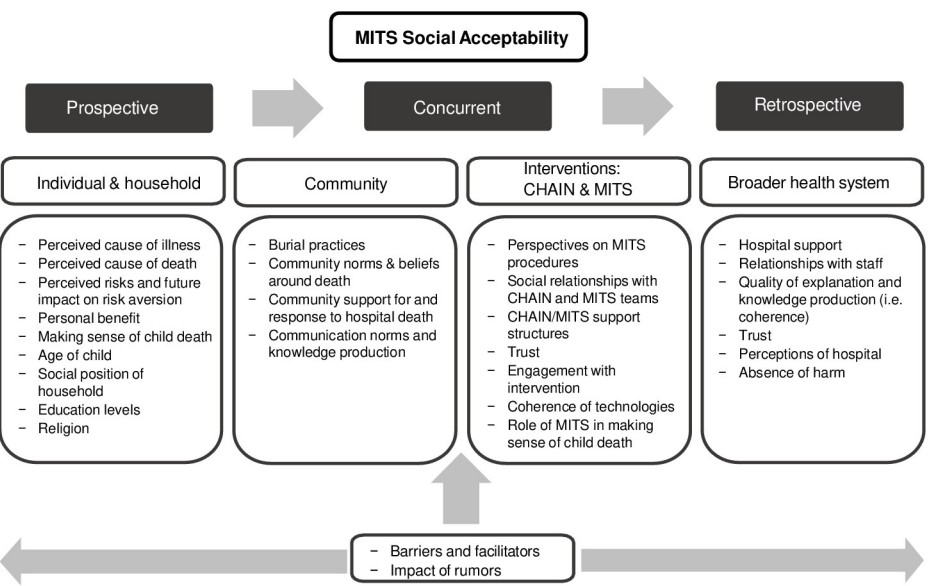

**Fig 1. MITS social acceptability conceptual framework.**

themes from interactions between codes. Second round parent discussion guides (see S6 Appendix) were informed by the first round of FGDs, and further explored relationships between social relations and MITS acceptability–an emergent theme. We developed a socio-ecological conceptual framework from common themes in the existing literature and emerging themes to further guide our analysis after the first round of FGDs (Fig 1). This framework highlights the potential barriers and facilitating factors to acceptability at the individual/house-hold, community, health system and intervention (i.e., MITS in this study) levels.

## Ethics approval

The Malawi National Health Sciences Research Committee (Protocol #17/09/1913) and the Oxford Tropical Research Ethics Committee (Reference 34–16) approved the study protocols. The University of Washington Institutional Review Board (STUDY00003689) exempted the study from review. Written informed consent was provided by all participants.

## Results

Participants across all groups expressed concerns about MITS but also recognized potential value for parents and improving child health. Attitudes toward MITS acceptability were inseparable from social context. Underlying social drivers included power inequalities in the home and hospital, and trusting relationships, or lack thereof, between parents and HCWs. Parents did not distinguish between HCWs and clinical researchers, identifying all as HCWs. Religious leaders did not notably differ in their responses. As such, we present their results together here rather than distinguishing between faith groups.

### Social relationships

Participants across FGDs emphasized that acceptability of MITS, especially in hospital settings, is tied to social relationships between parents and extended family members and with HCWs. Social relationships between family members often revolved around power to consent, whereas relationships between families and HCWs more often focused on the extreme power differential in hospital settings and limited communication that reduced families' trust of HCWs. Central sub-themes that informed social relationship dynamics included power and trust, and communication as a tool to navigate power relations and develop trust.

### Power and trust within the household

Family power structures were considered important influencers of MITS acceptability and were often reported as contingent on and determined by gender, age, wider community norms, and factors linked to income-earning or relative social position. The role of extended family in healthcare-related decision-making, including MITS, was perceived to be influential if they reside proximate to parents or if parents adhere to traditional decision-making roles.

Maternal uncles are often regarded as the head of family lineage and frequently control decisions regarding children within the family in traditional, matrilineal systems found across some Malawian ethnic groups, such as the Chewa of southern Malawi. However, some participants felt that evolving social norms have modernized these traditional structures. Many participants felt that primary responsibility for MITS consent decisions resides with parents:

> . . .so, we as the owners in pain of the loss of the child have the authority to know what has happened and [can accept] the child's body should be tested.
>
> (Participant 9, Fathers Round 2).

However, others feared that blame might be attached to parents who make autonomous decisions without consulting extended family members:

> The relatives from both sides need to be involved because if it's just two of you [parents], they will say that 'you are doing that on your own, then there must be something that you know [about the child's death]'. . ..
>
> (Participant 2, Mothers Round 1).

Participants also identified gender-based power norms as determinants of consent to research. Most felt that fathers would ultimately determine consent. While mothers are usually the primary carers in hospitals, mother FDG participants rarely felt they possess the power to consent to research or clinical procedures independently and researchers echoed this:

*The mothers are always reluctant that 'I cannot consent by myself to join the study when my husband doesn't know about it.'*

(Participant 5, CHAIN Team).

The majority of participants felt that MITS consent would be hindered if mothers are approached independently and fathers are difficult to reach.

## Power and trust within the healthcare system

Participants also described the importance of relationships in healthcare settings as an important dimension of research acceptability. The medical knowledge gap between HCWs and the community was emphasized by religious leaders, hospital support staff, and parents to intensify feelings of disempowerment experienced by patients and families during interactions with HCWs:

*. . . the hospital personnel take themselves as knowledgeable people, very educated, so they take anyone who goes there as an illiterate person, as a fool.*

(Participant 4, Religious Leaders).

This assumption of HCW monopoly over knowledge was reinforced by hospital support staff, who reported that HCWs enforce their power over patients or guardians by restricted information-sharing:

*. . .they just write in files, they do not say what operation we have done, this is what we will do, with admission for this number of days* [how long the child might be hospitalized following a medical procedure].

(Participant 8, Hospital Support Staff).

Religious leaders also highlighted that patients and families do not typically ask questions about diagnoses, treatment, or CoD, nor are they encouraged to do so, and this lack of expressed agency appears to lead to fatalistic acceptance following a negative hospital outcome, including a death:

*. . .once you have taken the person to the hospital, I have never seen anyone asking how the person has died; you just hear that the person who went to the hospital is dead and you accept it.*

(Participant 3, Religious Leaders).

Mistrust of HCWs and the healthcare system was considered foundational to rumors that HCWs might purposely kill children for blood stealing, organ harvesting, or Satanic practices (i.e. in the Malawian context, mysterious actions done for financial gain, often accompanied by physical harm). Parents and religious leaders highlighted existing rumors associated with routine hospital processes post-mortem, specifically holding the body in the morgue, washing of the body by mortuary staff, and organ harvesting to make medicines.

*She will not accept it [MITS] because when a person dies, they immediately take the body to the mortuary, they may conduct an autopsy without you knowing.* (Participant 4, Mothers Round 2) *They say that you should not bathe the body, so you will not know.*

(Participant 8, Mothers Round 2).

HCWs and the CHAIN team also emphasized that parent and other community members' hospital experiences prior to death are likely to inform trust and subsequent response to MITS. For example, families are often distrustful of blood sampling, especially when volumes appear excessive, whether for research or clinical purposes.

Less extreme situations involving disrespectful communication or inattentiveness by hospital staff were also cited as causes of mistrust toward HCWs, and occasionally, hospital support staff:

*There are some doctors at the hospital, and we have seen this at [QECH], you arrive, and it is like you are not even there.*

(Participant 8, Fathers Round 2).

## Communication as a bridge to trust

Open and respectful communication, where HCWs share regular, informative updates about the child's health status and respond to parental questions, was described by parents and religious leaders as integral to relationship building. These FDG participants also recommended that such communication begin from first encounters at the hospital. However, HCWs highlighted that resource constraints, such as insufficient staffing and limited time to attend to patient needs, hamper their communication and ability to foster trust with parents.

Hospital support staff, such as cleaners and patient attendants, reported they more easily build trust with families than HCWs, since they are from similar socioeconomic backgrounds:

*. . .they [parents] mostly fear the doctor and nurses. . . and us with the non-clinical positions, we chat with them and know their secrets. . ..*

(Participant 4, Hospital Support Staff).

Hospital support staff believed parents' social distance with and limited trust of HCWs would limit MITS acceptability. Mothers generally confirmed the sentiment that hospital support staff are more compassionate towards them than HCWs. However, this opinion was not shared by most fathers:

*[Hospital support staff] are the ones ruining things, if there is rudeness in the hospitals, they are the perpetrators.*

(Participant 2, Fathers Round 2).

Another father in the same FGD concurred and noted hospital support staff are inconsiderate, especially in their enforcement of rules concerning visiting hours or access to the wards.

Irrespective of the type of hospital personnel (HCW or support staff) referenced, both religious leaders and parents emphasized the need for mutual respect and open, thorough communication in order to foster trust and stressed these as critical for MITS acceptability.

*. . .there is a need for a good relationship and explanations. . .you should sit down with the person and explain to them clearly and the person should understand; with a good relationship, the person will accept. . . there should be explanations and not commands.*

(Participant 3, Fathers Round 2).

## Perceived benefits and risks of the MITS procedure

While social relationship dynamics and communication strategies were considered dominant potential influencers of MITS acceptability, participants also described procedure-associated benefits and risks (Table 2).

**Potential benefits of MITS.**   The potential of MITS to provide additional, more accurate information about CoD was emphasized in all FGDs as the most important benefit, and was considered especially relevant for families of children who died in the community because

**Table 2. Benefits and risks associated with MITS.**

| Benefits | Focus groups noting the concept | MITS in Malawi approach following formative research |
|---|---|---|
| Potential for additional CoD information to aid in sense-making of childhood death | HCWs | Study will support families after death with emotional support 4–6 weeks post-death and provide MITS findings |
| | CHAIN team | |
| | Hospital support staff | |
| | Mothers (Rounds 1,2) | |
| | Fathers (Rounds 1,2) | |
| | Religious leaders | |
| CoD information could be protective for family and community members | HCWs | Information provided on initial CoD 4–6 weeks post-death |
| | CHAIN team | |
| | Mothers (Rounds 1,2) | |
| | Fathers (Rounds 1,2) | |
| | Religious leaders | |
| Reduce accusations of witchcraft in community | Mothers (Rounds 1,2) | Information provided on initial CoD 4–6 weeks post-death |
| | Fathers (Round 2) | |
| Knowledge gained from MITS procedures could benefit the wider community (e.g., protect children from preventable illnesses and deaths through sharing information and recommendations more broadly within communities, discovery of new diagnoses and interventions) | HCWs | Information provided on initial CoD 4–6 weeks post-death. |
| | CHAIN team | |
| | Mothers (Rounds 1,2) | |
| | Fathers (Rounds 1,2) | Families encouraged to share findings within their social networks, if they desire. Plans for dissemination of study results (without identifiable information) broadly |
| | Religious leaders | |
| **Risks** | **Focus groups noting the concept** | **MITS in Malawi approach following formative research** |
| Disfigurement of the body | HCWs | Minimally invasive procedures described in consent process and families provided opportunity to view the body post-MITS |
| | Hospital support staff | |
| | Mothers (Round 2) | |
| | Fathers (Round 2) | |
| | Religious leaders | |
| Organ and blood harvesting | Hospital support staff | Foster trust with CHAIN team |
| | Mothers (Rounds 1,2) | |
| | Fathers (Round 2) | |
| | Religious leaders | |
| Disruption to transportation and burial plans | HCWs | Offer transportation and coffin* |
| | Religious leaders | |
| Potential identification of stigmatized illnesses that could cause challenges to families if revealed to community | Mothers (Round 1) | Strict confidentiality. Results to be provided to family members who consented. In instances where vertical transmission of a stigmatized illness is identified, every effort will be made to first discuss with the mother privately |
| Fuel rumors that Satanic practices occur at QECH | HCWs | Rumour surveillance system |
| | CHAIN team | |

* Transportation, coffin, and family support to be offered to all families approached for consent in MITS, not just those who accept to participate.

they would be less likely to have a medical diagnosis that could be interpreted as the CoD. Others felt parents would want to know what caused a child to die, irrespective of the location:

> *It's very important because most of the times if [the child dies under hospital care] we accept that the child is dead, but we still want to know what has killed him.*

(Participant 3, Mothers Round 1).

Parents and HCWs thought that identification and knowledge of hereditary illnesses would be particularly valuable for families. They also acknowledged that MITS might benefit their broader communities through the identification of new diagnoses and medical breakthroughs.

**Potential risks of MITS.** Participants in different FGDs highlighted different risks, with much more variability within and across groups compared to the potential benefits of MITS. The primary perceived risks of MITS were body disfigurement and fears of organ and blood harvesting for use in hospital procedures:

> *. . .when a child dies suddenly, (people say) there is something they wanted to get. . . like the kidneys, the liver, anything that is possible to transplant. . ..*

(Participant 8, Fathers Round 2).

While HCWs noted these risks, they also highlighted MITS could fuel rumors of Satanic practices at QECH:

> *People out. . .already say, 'there are Satanists there. . .we have heard that when they take the body to the mortuary, they tend to take the body parts.' Are they not going to say I think HCWs at QECH are now coming to an open with their Satanism practices? They were telling us they want to take body parts. . ..*

(Participant 5, HCWs).

### Reflections on appropriate forms of approach to families

Participants described various ways to sensitize the community about MITS and strategies for approaching parents for consent—from community-wide sensitization and media campaigns to informing an individual family only after a child death (Table 3). Engagement of extended family members was thought to be best determined by parents rather than assuming this outreach based on widespread community norms.

## Discussion

Our findings demonstrate that social relationships–between parents, family members, and hospital and research staff–are central to the hypothetical acceptability of MITS. Participants perceived unequal power relationships between HCWs and families as ubiquitous within hospitals and felt these could adversely influence MITS consent if families feel obligated to agree to the procedures, even if they find them unacceptable. This echoes findings by other investigators that consent given in an environment with unequal power cannot easily be equated with acceptability [35–37]. Hospital spaces are perceived as places in which ultimate power rests with hospital staff, given their medical knowledge and a perceived unwillingness to share that knowledge, while hospital rules, such as restricted visiting hours, reinforce parents' feelings of powerlessness [38–40]. Under such conditions, researchers should remain vigilant as to whether consent is driven by true acceptability or influenced by these underlying relations of power.

**Table 3. Reflections on appropriate forms of approach.**

| Approach factor | Focus groups noting the suggestion | Focus Group Suggestions |
|---|---|---|
| **Sensitization of broader community** | Hospital support staff Mothers (Round 1) Fathers (Round 1) | Civic education about MITS through community awareness campaigns |
| **Sensitization of parents/ families of children admitted to hospital** | Hospital support staff Mothers (Round 2) Fathers (Round 2) | Conduct sensitization with parents/families after hospital admission and before death |
| | HCWs | Avoid sensitization with parents/families in hospital before death to reduce the risk of parents/ families taking the child out of the hospital against medical advice |
| | CHAIN team | |
| **Introducing MITS to parents/ caregivers** | | |
| Who approaches | HCWs | Introduction to MITS study team by familiar HCW |
| | Mothers (Round 2) | |
| | Fathers (Round 2) | |
| Timing | HCWs | Families will need time to grieve—timing will depend on individual family needs |
| | CHAIN team | |
| | Mothers (Round 1) | |
| **Consent** | | |
| Information to share | HCWs | General information about MITS should be presented with the opportunity for potential participants to ask questions |
| | Hospital support staff | |
| | Fathers (Round 2) | |
| Who should be present | HCWs | Ideally, fathers will be present as they will have the ultimate authority to consent to MITS |
| | Hospital support staff | |
| | CHAIN team | |
| | Mothers (Rounds 1,2) | |
| | Fathers (Rounds 1, 2) | |
| | HCWs | In some instances, extended family members (especially maternal uncles) should be contacted |
| | Hospital support staff | |
| | Mothers (Rounds 1,2) | |
| | Fathers (Rounds 1,2) | |

Religious leaders, hospital support staff, and parents all identified open communication and establishing a sense of shared control in the care of the child as a way to address this power imbalance. Our results demonstrate that especially when offering a new and potentially controversial intervention in an emotionally difficult situation—i.e. MITS after the death of a child, trust in HCWs and open communication could deeply influence decision-making and acceptability.

Our findings expand on prior research demonstrating trust as a key factor shaping acceptability of treatments and care in health and research settings [15,41–45]. Prior MITS studies found that a positive health facility reputation and respectful care by HCWs contribute to MITS acceptability whilst mistrust of the health system, HCWs, and researchers act as barriers [6,24]. These findings and our own reflect the importance of social relationships on acceptability of medical and/or research procedures. Trust and power dynamics are likely to play an even greater role in hospital-based MITS studies that enroll participants who die during inpatient care. While prior experiences with the health care system and HCWs will likely play a role in MITS acceptability, data from this study suggest the immediacy of family experiences

with hospital and research staff during the index inpatient stay will likely be even more important drivers.

Respectful care during maternity services has recently received much attention [46,47]; a similar emphasis is needed throughout the healthcare system, including how families are treated during their child's engagement in health care and research. Of course, institutionalizing respectful care depends on many factors including incorporation into HCW and research staff training, strengthening health systems, enhancing HCW job satisfaction and fostering respect within the system.

Power dynamics and gender norms within households also emerged as important factors determining MITS acceptability. While mothers assume primary responsibility for care of their children during hospitalization, decision-making power over consent to procedures or engagement in research often resides with fathers. Participants stressed that MITS consent processes might be hindered if fathers are difficult to reach or find the procedures unacceptable. South African and Mozambican studies have also reported gender power relations and limited access to household decision-makers as dominant influencers of community and hospital-based MITS acceptability [25,26].

Interestingly, while previous research has found religion to influence acceptability of MITS compared to full autopsy, particularly in predominantly Muslim communities [6,24,48], religious leaders in this study highlighted social relationships with the health system as more influential to MITS acceptability. Ultimately, participants felt that individual religious beliefs play a larger role than broader norms within religious communities.

Our formative study has several limitations, the primary one being the hypothetical nature of the discussions. Only one of the participants had experienced the loss of a child, none had actually experienced being approached to discuss or consent to autopsy or MITS. Within Malawi, social norms and relationships, as well as burial and funeral rites, differ across contexts and these results might not be generalizable to all settings. Additionally, this study focused on deaths occurring during inpatient stays. Thus, caution must be exercised in extrapolating these findings to community-based MITS studies. However, the underlying issues of trust, power dynamics and the importance of communication are widely relevant to diverse settings. We recommend that future MITS studies in other settings also conduct formative research to explore these complex factors specific to their unique contexts.

Despite limitations, this formative research provided a foundation for implementation of a MITS study to elucidate CoD among hospitalized young Malawian children. Important lessons learned informed development of consent and study procedures. Furthermore, this formative study corroborates findings across a body of emerging literature on acceptability of MITS [6,24–26,49], a relatively new procedure. Our study adds knowledge about the centrality of social relationships to pediatric MITS acceptability, including the roles of trust, power, and communication within relationships. In order to improve patient understanding about MITS, and maximize MITS acceptability, especially within hospital-based settings, researchers and HCWs will need to focus on fostering supportive, respectful relationships, and open communication with patients and their families that is initiated well before the death of a child, commencing at presentation to hospital and throughout the inpatient stay. Only by taking this complex and multi-layered social environment into account, can decisions to consent to MITS in children be understood and true acceptability achieved.

## Supporting information

**S1 Appendix. Discussion guide: Frontline health care workers.**
(DOC)

**S2 Appendix. Discussion guide: Hospital support staff.**
(DOC)

**S3 Appendix. Discussion guide: CHAIN team.**
(DOC)

**S4 Appendix. Discussion guide: Parents round 1.**
(DOC)

**S5 Appendix. Discussion guide: Religious leaders.**
(DOC)

**S6 Appendix. Discussion guide: Parents round 2.**
(DOC)

## Acknowledgments

We would like to thank all participants for making this study possible as well as Hendrina Kaliati and Elvis Moyo of the Malawi Liverpool Wellcome Trust Clinical Research programme for helping with transcription and translation of recorded data and identifying parent participants, respectively. We thank the Principal Investigators of CHAIN, Prof. Judd L Walson and Prof. James A Berkley, for their support.

## Author Contributions

**Conceptualization:** Cornelius Huwa, Sassy Molyneux, Donna M. Denno, Nicola Desmond.

**Data curation:** Dave Namusanya, Andrew Hamuza, Dennis Chasweka.

**Formal analysis:** Sarah Lawrence, Dave Namusanya, Andrew Hamuza, Donna M. Denno, Nicola Desmond.

**Funding acquisition:** Wieger Voskuijl, Donna M. Denno, Nicola Desmond.

**Investigation:** Sarah Lawrence, Dave Namusanya, Donna M. Denno, Nicola Desmond.

**Methodology:** Dave Namusanya, Andrew Hamuza, Cornelius Huwa, Donna M. Denno, Nicola Desmond.

**Project administration:** Dennis Chasweka, Wieger Voskuijl, Donna M. Denno, Nicola Desmond.

**Supervision:** Cornelius Huwa, Wieger Voskuijl, Donna M. Denno, Nicola Desmond.

**Writing – original draft:** Sarah Lawrence, Dave Namusanya.

**Writing – review & editing:** Sarah Lawrence, Dave Namusanya, Maureen Kelley, Sassy Molyneux, Wieger Voskuijl, Donna M. Denno, Nicola Desmond.

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
