## [Decision Letter · Decision Letter 0]

23 Oct 2020

PONE-D-20-25933

Hypothetical acceptability of hospital-based post-mortem pediatric minimally invasive tissue sampling in Malawi: The role of complex social relationships

PLOS ONE

Dear Dr. Lawrence,

Thank you for submitting your manuscript to PLOS ONE. After careful consideration, we feel that it has merit but does not fully meet PLOS ONE’s publication criteria as it currently stands. Therefore, we invite you to submit a revised version of the manuscript that addresses the points raised during the review process.

We look forward to receiving your revised manuscript.

Kind regards,

Ritesh G. Menezes, M.B.B.S., M.D., Diplomate N.B.

Academic Editor

PLOS ONE

2. Please include a copy of the interview guide used in the study, in both the original language and English, as Supporting Information, or include a citation if it has been published previously.

3.In your Data Availability statement, you have not specified where the minimal data set underlying the results described in your manuscript can be found. PLOS defines a study's minimal data set as the underlying data used to reach the conclusions drawn in the manuscript and any additional data required to replicate the reported study findings in their entirety. All PLOS journals require that the minimal data set be made fully available. For more information about our data policy, please see http://journals.plos.org/plosone/s/data-availability.

………………………………….

Additional Academic Editor Comments:

• Provide further details in relation to the availability of data.

• A sentence or two on the Childhood Acute Illness and Nutrition (CHAIN) Network should be provided in the ‘abstract’ section.

• The defined objectives or research questions of the present study should be detailed in the concluding paragraph of the ‘introduction’ section.

• Provide explicit details on the CHAIN Network in the ‘methods’ section.

• Methods-1st sentence: “The formative acceptability research was conducted in advance of the MITS in Malawi study, which was designed as a CHAIN substudy to assess CoD in patient deaths.” Provide further explanation in relation to this sentence so that the reader understands clearly the methods being considered in the present study.

• Methods-“Religious leaders affiliated with religions that prohibit medical care seeking were excluded”. Which were such religions encountered by the researchers of the present study?

• Methods: Provide additional details on the sampling strategy, including rationale for the recruitment method and participant inclusion/exclusion criteria.

• Methods: Provide data analysis procedures described in detail to enable replication.

• For details on reporting standards related to qualitative research refer to the webpage https://journals.plos.org/plosone/s/submission-guidelines#loc-qualitative-research while revising the manuscript.

• Discussion-1st sentence: Be specific to minimally invasive tissue sampling (MITS).

• Address all the comments made by the reviewers.

Reviewers' comments:

Reviewer's Responses to Questions

**Comments to the Author**

1. Is the manuscript technically sound, and do the data support the conclusions?

Reviewer #1: Yes

Reviewer #2: Yes

Reviewer #3: Yes

2. Has the statistical analysis been performed appropriately and rigorously? 

Reviewer #1: Yes

Reviewer #2: N/A

Reviewer #3: N/A

3. Have the authors made all data underlying the findings in their manuscript fully available?

Reviewer #1: Yes

Reviewer #2: Yes

Reviewer #3: Yes

4. Is the manuscript presented in an intelligible fashion and written in standard English?

Reviewer #1: Yes

Reviewer #2: Yes

Reviewer #3: Yes

5. Review Comments to the Author

Reviewer #1: The manuscript focuses on hypothetical acceptance of MITS in Malawi, which is planned to be conducted at a hospital. While the formative research has tried to focus and present the perspectives of various stakeholders, there are some comments/concerns that need attention.

1. Introduction:

1.1. Context- Please clarify which age is included in young children under the CHAIN study.

1.2. Hypothesis: We understand that the perspectives and perceptions of the different stakeholders are different for MITS. The study included only FGDs for data collection. The participants included had limited/no experience of child death. Why was interview with some parents who had child death not planned?

1.3. Line 97-98, "A few studies have been conducted to ascertain MITS acceptability in low resource settings, including in Sub-Saharan Africa (6, 15-17)."

It refers to including, but references are given primarily for African context. Is so, the statement should be modified. There are information available from other developing countries.

2. Methods

2.1. Study design: Topic guides in English should be added as Annexure.

2.2. Data collection: Table 1: Should add the age of the participants. Composition from different religions should be given. Who conducted the FGDs? Where the FGDs for non-HCW participants conducted?

3. Results.

3.1. Opinion of Religious leaders from their religion perspectives: What were the perspectives of the different religious leaders about MITS?

3.2. Was there any difference in perspectives of the different types of HCWs- doctors, nurses and support staffs?

Reviewer #2: This aligns with some existing research but provides an interesting perspective. While many hypothetical studies regarding the acceptability of MITS focus on religion as a barrier (in LMICs), this manuscript highlights trust and communication as a key factor. Any many MITS programs seek religious documentation or support from religious leaders, this manuscript suggests that there are other barriers to MITS that are not being fully addressed as part of MITS studies.

Specific comments include:

Line 104: It is not clear how this statement relates to the rest what is said previously. Is this the justification for this study?

Line 108: Excellent point about ‘acceptability’.

Line 118: how does this approach limit insight into acceptability?

Line 121 If there are limitations to understanding ‘acceptability’ within the construct of comparing one intervention over another, how was acceptability defined in this study?

Line 143 Where the CHAIN research team members not involved in the MITS in Malawi sub-study?

Line 190 Was the information gathered in the first round of FDGs with parents used to inform the interview guides for round 2? Were the FDGs for mothers separate from those FDGs including fathers?

Line 215: is it possible to say more about the social relationships? Was it an issues between HCWs?

Line 227 This sentence is awkward. Is there a word missing?

Line 247 Who is being referred to in the statement ‘rarely felt they possess the power to consent to research….’.Is this a generalization from all FGD participants or specifically from mothers?

Line 315: In what ways did you find that resources limit/constrain communication and fostering trust?

Line 371: This sentence is confusing.

Line 378: Was the fear of body disfigurement primarily with mothers? Fathers? Both?

Line 433-436: It is not clear to me how this conclusion was drawn based on what was shared in the manuscript

Reviewer #3: This is a very interesting article that explores the hypothetical acceptability of hospital-based post-mortem paediatric minimally invasive tissue sampling in Malawi. This article contributes to the scarce knowledge on acceptability of MITS using socio-anthropological methods and qualitative data, mainly focusing, among other aspects, on relations among the different actors involved or influencing decision-making and power relations. I congratulate the authors for the study and for the article. All the sections are clearly explained, the manuscript is well written in standard English and the language used is also clear and understandable. I recommend to add a paragraph describing the study area, as it will be very helpful for readers unfamiliar with the context of Blantyre and Malawi. I think is also necessary to further explain the conceptual framework at the end of the Methods section, describing the themes and the associations. I recommend accepting the article for publication after addressing these suggestions and the minor comments and sugestions explained below.

ABSTRACT

Line 51: I would add that community members include religious leaders and parents. i.e.: “We conducted eight focus group discussions with key hospital staff and community members (including religious leaders and parents of children under 5) to explore attitudes towards…”

INTRODUCTION

Line 85: The authors explain that “MITS in Malawi also employs novel post mortem endoscopic gastrointestinal sampling…” but the article is about acceptability prior implementation. When MITS started in Blantyre? As it reads now, it may seem that it was during the qualitative study. I would add when MITS started to be performed.

Line 91: Replace “autopsy” for “complete diagnostic autopsy” (CDA)

STUDY DESIGN AND SAMPLING

Study site: Consider to add a paragraph describing the study area as it would be useful for the reader to contextualize the study. For example, it will be useful to know the main existing religions, ethnic groups, main sources of income…

Line 139: I think there is a mistake in the reference 16, when talking about researchers from Mozambique that had been interviewed, as reference 16 is a very specific part of their work. In this paper there is a paragraph on the main study design, but to refer to it in a broader sense, I would suggest to cite instead: REFs: 9. Bassat Q, Castillo P, Martinez MJ, Jordao D, Lovane L, Hurtado JC, et al. Validity of a minimally invasive autopsy tool for cause of death determination in pediatric deaths in Mozambique: An observational study. PLOS Med. 2017;14(6):1549-676 and Maixenchs M, Anselmo R, Martínez Pérez G, Oruko K et al. (2019) Socio-anthropological methods to study the feasibility and acceptability of the minimally invasive autopsy from the perspective of local communities: lessons learnt from a large multi-centre study, Global Health Action, 12:1, 1559496, DOI: 10.1080/16549716.2018.1559496

DATA COLLECTION AND ANALYSIS

Line 186: Add (QSR International Pty Ltd) after Nvivo 11 Plus; instead of the reference (31).

Line 196: As I previously explained, the figure of the conceptual framework is quite confusing. Please introduce a paragraph explaining it.

ETHICS APPROVAL

Line 199: Include the reference number for the approvals (Protocol #17/09/1913, Reference 34-16, STUDY00003689), as you did in the Ethics Statement.

RESULTS:

Line 229: Include which Malawian ethnic groups.

Line 242: Quote. When the participant said that “(…) then there must be something that you know [about the child´s death]“ is she refereeing to something that the parents want to hide? As issues related with stigmatized diseases or mothers not taking care properly of their child? If yes, specify in line 237, where you are explaining blame.

Line 272: I do not understand the quote, when it reads “(…) with admission of this number of days”. Please, clarify.

Line 285: Very interesting. It would be good to describe in more detail which are the ordinary procedures explained by participants when someone dies at the hospital and who is responsible/does each one. For example: Who certifies the death? Who washes the body? Does it have a cost? Is the family allowed to wash the body themselves? How long since the person dies and the body is returned to the family?

Line 315: Which kind of resource constraints limit communication between HCW and parents? Are the authors referring to lack of staff, so they are overwhelmed and cannot attend the parents? About lack of a proper space at the hospital where HCW can talk with the parents privately and calmly? About no resources to improve personnel skills? Please, explain.

Line 322: Clarify in the quote who were “they”: “…they [parents? patients?] fear the doctor and nurses (…)

Line 329: Specify if all fathers in the FGD, almost all fathers, some or few.

Line 254, table 2: Benefits and risks associated with MITS.

- Which was the MITS in Malawi approach following formative research (column 3) for “Knowledge gained from MITS procedures could benefit the wider community” (column 1, row 5), if one?

- In the row “Disfigurement of the body” (column 1), the MITS in Malawi approach following formative research (column 3) is “Minimally Invasive Procedure”. Please explain. Did the protocol on how to perform MITS changed after this finding?

- In the row “Potential identification of stigmatized illnesses that could cause challenges to families if revealed to community” (column 1), the MITS in Malawi approach following formative research was “strict confidentiality” (column 3). Please specify, both regarding the MITS´ results delivery strategy and regarding who are the only ones receiving the results. Parents? The ones that the parents have previously decided?

- Line 356: * Transportation, coffin, and family support offered to all families approached for consent in MITS, not just those who participated. I would say instead: *Transportation, coffin, and family support to be offered to all families approached for consent in MITS, not just those who accept to participate.

Line 402: Table 3: Reflections on appropriate forms of approach.

- I would suggest table 2 and 3 to maintain the same order for the columns (“Focus group noting the concept/suggestion” is placed in column 2 in the table 2 and in column 3 in table 3).

- I do not understand the text “Wait to sensitize until after hospital admission or risk parents leaving against medical advice” (column 2, row 3). Please, clarify.

- I would replace “[Avoid sensitization with] Risk parents leaving against medical advice” for “parents with perceived risk of them taking the child out of the hospital against medical advice” or similar. Is this what authors want to say?

- “Complex and detailed MITS procedure information will not translate well in Chichewa and might worry participants” (Approach factor. Consent, column 2): This is not a suggestion.

DISCUSSION:

Line 479. I would replace “Our study adds data…” for “Our study adds knowledge…”.

REFERENCES:

Line 499: Reference 2: Add when data were retreived: 2. UNICEF. Under-five mortality: Child mortality data 2019 [Available from: https://data.unicef.org/topic/child-survival/under-five-mortality/. Accessed on XXX]

Line 579: Delete reference 31.

6. PLOS authors have the option to publish the peer review history of their article (what does this mean?). If published, this will include your full peer review and any attached files.

Reviewer #1: **Yes: **Manoja Kumar Das

Reviewer #2: No

Reviewer #3: No

---

## [Author Response · Author response to Decision Letter 0]

6 Dec 2020

Editor comments and responses:

-Provide further details in relation to the availability of data.

The underlying data is available on Figshare and the DOIs are listed below.

https://doi.org/10.6084/m9.figshare.13335365.v1

https://doi.org/10.6084/m9.figshare.13335425.v1

https://doi.org/10.6084/m9.figshare.13337567.v1

https://doi.org/10.6084/m9.figshare.13338467.v1

https://doi.org/10.6084/m9.figshare.13337615.v1

https://doi.org/10.6084/m9.figshare.13337612.v1

https://doi.org/10.6084/m9.figshare.13338632.v1

https://doi.org/10.6084/m9.figshare.13338635.v1

- A sentence or two on the Childhood Acute Illness and Nutrition (CHAIN) Network should be provided in the ‘abstract’ section.

We have added details on CHAIN on p 3 lines 50-55 which reads “‘MITS in Malawi’ is a sub-study of the Childhood Acute Illness & Nutrition (CHAIN) Network, which aims to identify biological and socioeconomic mortality risk factors among young children hospitalized for acute illness or undernutrition. MITS in Malawi employs standard MITS and a novel post-mortem endoscopic intestinal sampling approach to better understand CoD among children with acute illness and/or malnutrition who die during hospitalization.”

-The defined objectives or research questions of the present study should be detailed in the concluding paragraph of the ‘introduction’ section.

We have restructured the introduction to strengthen our justification for this research and now specify the research objections on p 7 lines 151-160. This now reads “In this study we defined acceptability as both an ethically and socially embedded construct that includes absence of harm, positive effect, personal benefit and anticipated perceptions of usefulness and framed this within the social, cultural, and religious context of Southern Malawi. This qualitative study aimed to assess the hypothetical acceptability of MITS and factors informing acceptability in the Malawian context in order to inform the design of optimal ethical strategies for consent-seeking and other study processes prior to commencement of the MITS in Malawi study. 

-Provide explicit details on the CHAIN Network in the ‘methods’ section.

We have added additional details on the CHAIN Network in the introduction section where it is introduced on p 5 lines 92, 94-95 that now reads “The Childhood Acute Illness & Nutrition (CHAIN) Network aims to identify biological and socioeconomic mortality risk factors among young children (2-23 months) hospitalized for acute illness or undernutrition across nine sites in six countries. Children with moderate and severe wasting or kwashiorkor are purposefully oversampled in order to enhance the capacity to examine this vulnerability.” We have also added a reference the CHAIN methods paper in our methods section for easier reference for readers.

-Methods-1st sentence: “The formative acceptability research was conducted in advance of the MITS in Malawi study, which was designed as a CHAIN substudy to assess CoD in patient deaths.” Provide further explanation in relation to this sentence so that the reader understands clearly the methods being considered in the present study.

We deleted this sentence from the study setting section and provided details about the methods in the study design and sampling section on p 9 lines 178-183. This section now reads “We utilized focus group discussions (FGD) to understand MITS acceptability and influencing factors. We reviewed the MITS consent and acceptability literature and interviewed researchers experienced in conducting pediatric autopsy and MITS in Mozambique and pediatric autopsy and minimally invasive brain sampling in Malawi to develop FGD topic guides in English, which were translated to Chichewa (the local language).”

- Methods-“Religious leaders affiliated with religions that prohibit medical care seeking were excluded”. Which were such religions encountered by the researchers of the present study?

Some apostolic churches prohibit medical care seeking. We have clarified the text on p 10 line 207 to specify the religious group, which now reads “Religious leaders affiliated with apostolic churches that prohibit medical care seeking were excluded.”

- Methods: Provide additional details on the sampling strategy, including rationale for the recruitment method and participant inclusion/exclusion criteria.

Thank you for this request, we have added additional details regarding the sampling strategies on p 9 lines 185-189 that now reads “All FGD participants were purposively sampled to capture a wide variety of perspectives regarding MITS acceptability across strata. FGD participants were selected through local leaders, all of whom have pre-existing relationships with Malawi-Liverpool-Wellcome Trust through the communication of science department” and p 9 lines 195-199 which reads “We included physicians (generalists and trainees), clinical officers, and nurses (including senior nurses) in the HCW FGD, guards, cleaners, patient attendants in the hospital support staff FGD, and the CHAIN study coordinator, study team clinicians, and fieldworkers in the CHAIN FGD. CHAIN and CHAIN Malawi site principal investigators were excluded from the CHAIN FGD.” 

-Methods: Provide data analysis procedures described in detail to enable replication.

We have expanded on data analysis procedures on p 12 lines 241-252, which now reads “Our initial codebook was developed from an open coding process derived from the data. Additions were made as the analysis proceeded. All transcripts were independently coded by two of three coders (AH, DN, SL) and later merged. Thematic content analysis was used to iteratively develop themes from interactions between codes. Second round parent discussion guides (see S6 Appendix) were informed by the first round of FGDs, and further explored relationships between social relations and MITS acceptability–an emergent theme. We developed a socio-ecological conceptual framework from common themes in the existing literature and emerging themes to further guide our analysis after the first round of FGDs (Figure 1). This framework highlights the potential barriers and facilitating factors to acceptability at the individual/household, community, health system and intervention (i.e., MITS in this study) levels.”

- For details on reporting standards related to qualitative research refer to the webpage https://journals.plos.org/plosone/s/submission-guidelines#loc-qualitative-research while revising the manuscript.

Thank you for providing this reference. We have utilized the cited webpage in addressing earlier comments to strengthen our methods section.

-Discussion-1st sentence: Be specific to minimally invasive tissue sampling (MITS).

We have modified the sentence to be specific to MITS on p 25 line 469-471. This now reads “Our findings demonstrate that social relationships–between parents, family members, and hospital and research staff–are central to the hypothetical acceptability of MITS.”

- Address all the comments made by the reviewers.

Thank you, we have addressed all reviewer comments as detailed below.

Reviewer 1 comments and responses: 

1.1. Context- Please clarify which age is included in young children under the CHAIN study.

We have now noted the age range on p 5 line 92 as between 2–23 months which now reads “The Childhood Acute Illness & Nutrition (CHAIN) Network aims to identify biological and socioeconomic mortality risk factors among young children (2-23 months) hospitalized for acute illness or undernutrition across nine sites in six countries.”

1.2. Hypothesis: We understand that the perspectives and perceptions of the different stakeholders are different for MITS. The study included only FGDs for data collection. The participants included had limited/no experience of child death. Why was interview with some parents who had child death not planned?

Thank you for this important question. We did not specifically interview parents who had lost a child due to ethical considerations. We were concerned this might be triggering for families and did not have the capacity to offer formal or trauma-informed grief counseling or emotional support during this formative research.

1.3. Line 97-98, "A few studies have been conducted to ascertain MITS acceptability in low resource settings, including in Sub-Saharan Africa (6, 15-17)."

It refers to including, but references are given primarily for African context. Is so, the statement should be modified. There are information available from other developing countries.

We have added the following citations for research conducted in other settings (Pakistan and India) on p 6 line 127. 

27. Feroz A, Noor Ibrahim M, McClure EM, Shiraz Ali A, Sunder Tikmani S, Reza S, et al. Perceptions of parents and religious leaders regarding minimal invasive tissue sampling to identify the cause of death in stillbirths and neonates: results from a qualitative study. Reproductive Health. 2019;16(53).

28. Das MK, Arora NK, Rasaily R, Kaur G, Malik P, Kumari M, et al. Perceptions of the healthcare providers regarding acceptability and conduct of minimal invasive tissue sampling (MITS) to identify the cause of death in under-five deaths and stillbirths in North India: a qualitative study. BMC Health Services Research. 2020;20(833).

29. Feroz A, Shiraz Ali A, Noor Ibrahim M, McClure EM, Sunder Tikmani S, Reza S, et al. Perceptions of health professionals regarding minimally invasive tissue sampling (MITS) to identify the cause of death in stillbirths and neonates: results from a qualitative study. Maternal Health, Neonatolgy, and Perinatology. 2019;5(17).

2.1. Study design: Topic guides in English should be added as Annexure.

We have added the topic guides as appendices S1-S6.

2.2. Data collection: Table 1: Should add the age of the participants. Composition from different religions should be given. Who conducted the FGDs? Where the FGDs for non-HCW participants conducted?

Thank you for these comments. We did not collect participant ages because all participants were adults and we did not believe age was a central factor to our questions of interest. We did not collect data on religious affiliations by participants other than religious leaders.

We have added on p 11 line 226 that a Malawian social scientist (co-author) conducted the interviews. 

On p 10 lines 203 and 210, we note that FGDs with non-HCW participants (religious leaders and parents) were conducted at nearby health centres rather than the hospital. These sections read “FGDs with religious leaders from prevalent religions in the Blantyre area (Adventist, Catholic, Pentecostal, and Presbyterian Christian denominations and Islam) were conducted at health centres” and “FGDs with mothers and fathers were held separately in health centres in two phases.”

3.1. Opinion of Religious leaders from their religion perspectives: What were the perspectives of the different religious leaders about MITS?

Religious leaders across faith groups did not notably differ in their beliefs about MITS. As such, we presented our results as attributable to the religious leaders as a whole, rather than by specific faith groups. We have added a note to clarify this on p 13 lines 269-271. This reads “Religious leaders did not notably differ in their responses. As such, we present their results together here rather than distinguishing between faith groups.”

3.2. Was there any difference in perspectives of the different types of HCWs- doctors, nurses and support staffs?

Differences between types of HCWs and hospital support staff cadres were not notable. However, some differences between groups (HCW vs. hospital support staff vs CHAIN team) were notable and are detailed in Tables 2 and 3 (pp20-21, 23-25). Of note, HCWs and the CHAIN team were worried that MITS would fuel rumours of their involvement in Satanic practices, whereas hospital support staff thought the families would identify organ harvesting and blood stealing as potential risks of participation. Their recommendations for sensitization differed because of this. HCWs and the CHAIN team suggested avoiding sensitization with families prior to the child’s death to avoid them taking the child out of the hospital against medical advice. In contrast, hospital support staff thought families should be sensitized to MITS upon admission to foster trust and ease fears of organ harvesting and blood stealing. 

Reviewer 2 comments and responses: 

Line 104: It is not clear how this statement relates to the rest what is said previously. Is this the justification for this study?

Thank you for pointing this out, it is part of the justification of the study. We have restructured the introduction and reworded this sentence on p 6 lines 131-133 which now reads” Given context-specific findings in past literature and known beliefs and practices surrounding child-raising and death in Malawi, it would have been risky to assume MITS acceptability in the Malawian context prior to implementation.”

Line 108: Excellent point about ‘acceptability’.

Thank you for this comment.

Line 118: how does this approach limit insight into acceptability?

By drawing largely on quantitative survey data this approach limits more detailed responses and nuanced information about acceptability. We have updated the text on p 6 line 123-125 to reflect this limitation which now reads “While valuable, this approach limits insight into acceptability of this novel procedure since it draws largely on quantitative survey data.”

Line 121 If there are limitations to understanding ‘acceptability’ within the construct of comparing one intervention over another, how was acceptability defined in this study?

We defined acceptability as “In this study we defined acceptability as both an ethically and socially embedded construct that includes absence of harm, positive effect, personal benefit and anticipated perceptions of usefulness and framed this within the social, cultural, and religious context of Southern Malawi,” which has been added to the text on p 7 lines 151-156.

Line 143 Where the CHAIN research team members not involved in the MITS in Malawi sub-study?

Thank you for this question. This formative research predated the launch of the MITS in Malawi study which did include some CHAIN team members, including the study coordinator, lab personnel, drivers, and one study clinician. The CHAIN study coordinator, study team clinicians, and fieldworkers were interviewed to understand their perspectives on MITS, which is now noted on p 9 lines 197-199 and now reads “We included… and the CHAIN study coordinator, study team clinicians, and fieldworkers in the CHAIN FGD. CHAIN and CHAIN Malawi site principal investigators were excluded from the CHAIN FGD.”

Line 190 Was the information gathered in the first round of FDGs with parents used to inform the interview guides for round 2? 

Second round discussion guides with parents were informed by the first round of FGDs. We have added this information on p 10 lines 225-226. 

-Were the FDGs for mothers separate from those FDGs including fathers?

FGDs for mothers were separate from those with fathers. On p 8 line 190, we note that “FGDs with mothers and fathers were held separately in health centres in two phases.”

Line 215: is it possible to say more about the social relationships? Was it an issues between HCWs?

We have expanded this introductory section of our results on social relationships on pp 12 lines 246 which now reads “Second round parent discussion guides (see S6 Appendix) were informed by the first round of FGDs, and further explored relationships between social relations and MITS acceptability–an emergent theme.”

Line 227 This sentence is awkward. Is there a word missing?

Thank you for pointing this out. We have added an extra word to clarify this sentence’s meaning on p 14 line 293. It now reads “Maternal uncles are often regarded as the head of family lineage and frequently control decisions regarding children within the family in traditional, matrilineal systems found across some Malawian ethic groups, such as the Chewa of southern Malawi.”

Line 247 Who is being referred to in the statement ‘rarely felt they possess the power to consent to research….’.Is this a generalization from all FGD participants or specifically from mothers?

Mothers rarely felt they had the power to consent to research. We have clarified this sentence by replacing “they” with “mother FGD participants” on p 15 line 312. 

Line 315: In what ways did you find that resources limit/constrain communication and fostering trust?

We have specified on p 18 lines 381-383 that specific resource constraints identified by HCWs included insufficient staffing and limited time to attend to patient needs. The text now reads “However, HCWs highlighted that resource constraints, such as insufficient staffing and limited time to attend to patient needs, hamper their communication and ability to foster trust with parents.”

Line 371: This sentence is confusing.

Thank you for noting this. We have reworded this sentence on p 22 lines 436-437 for clarity. It now reads “Parents and HCWs thought that identification and knowledge of hereditary illnesses would be particularly valuable for families.”

Line 378: Was the fear of body disfigurement primarily with mothers? Fathers? Both?

Thank you for this question. In Table 2, Row 7, we identify this perceived risk as shared by HCWs, hospital support staff, mothers (round 2) and fathers (round 2) and religious leaders.

Line 433-436: It is not clear to me how this conclusion was drawn based on what was shared in the manuscript

Thank you for this comment. We drew this from conclusion from the fathers’ discussion of inattentiveness and rudeness experienced at QECH on p 18 lines 373-374 which reads “There are some doctors at the hospital, and we have seen this at [QECH], you arrive, and it is like you are not even there. (Participant 8, Fathers Round 2) and on p 19 lines 398-399 which reads “[Hospital support staff] are the ones ruining things, if there is rudeness in the hospitals, they are the perpetrators. (Participant 2, Fathers Round 2).”

Reviewer 3 comments and responses:

ABSTRACT

Line 51: I would add that community members include religious leaders and parents. i.e.: “We conducted eight focus group discussions with key hospital staff and community members (including religious leaders and parents of children under 5) to explore attitudes towards…”

We have added this information in the abstract on p 3 line 62 which now reads “We conducted eight focus group discussions with key hospital staff and community members (religious leaders and parents of children under 5) to explore attitudes towards MITS and inform consent processes prior to commencing the MITS in Malawi study.”

INTRODUCTION

Line 85: The authors explain that “MITS in Malawi also employs novel post mortem endoscopic gastrointestinal sampling…” but the article is about acceptability prior implementation. When MITS started in Blantyre? As it reads now, it may seem that it was during the qualitative study. I would add when MITS started to be performed.

We have updated p 5 line 999 to reflect this detail by changing the language to be in the future tense which now reads “MITS in Malawi also will also employ novel post-mortem endoscopic gastrointestinal sampling to assess the contribution of gastrointestinal pathology to mortality among acutely ill and undernourished children.”

Line 91: Replace “autopsy” for “complete diagnostic autopsy” (CDA)

We have replaced autopsy with CDA as suggested on p 5 line 105.

STUDY DESIGN AND SAMPLING

Study site: Consider to add a paragraph describing the study area as it would be useful for the reader to contextualize the study. For example, it will be useful to know the main existing religions, ethnic groups, main sources of income…

We added study area details about the population with the catchment area on p 8 lines 170-155. This text reads “Eligibility for CHAIN and the MITS in Malawi study included residence within a 50-kilometre radius of QECH, encompassing both urban and rural communities, including high density urban informal settlements and surrounding agricultural areas linked by high rates of mobility and income-driven migration. Southern Malawi is increasingly ethnically diverse but dominated by matrilocal ethnic groups such as the Chewa who are largely Christian.”

Line 139: I think there is a mistake in the reference 16, when talking about researchers from Mozambique that had been interviewed, as reference 16 is a very specific part of their work. In this paper there is a paragraph on the main study design, but to refer to it in a broader sense, I would suggest to cite instead: REFs: 9. Bassat Q, Castillo P, Martinez MJ, Jordao D, Lovane L, Hurtado JC, et al. Validity of a minimally invasive autopsy tool for cause of death determination in pediatric deaths in Mozambique: An observational study. PLOS Med. 2017;14(6):1549-676 and Maixenchs M, Anselmo R, Martínez Pérez G, Oruko K et al. (2019) Socio-anthropological methods to study the feasibility and acceptability of the minimally invasive autopsy from the perspective of local communities: lessons learnt from a large multi-centre study, Global Health Action, 12:1, 1559496, DOI: 10.1080/16549716.2018.1559496

Thank you, we have updated the citations as suggested on p 9 line 181.

DATA COLLECTION AND ANALYSIS

Line 186: Add (QSR International Pty Ltd) after Nvivo 11 Plus; instead of the reference (31).

We have added “QSR International Pty Ltd” on p 12 line 241 and removed reference 31.

Line 196: As I previously explained, the figure of the conceptual framework is quite confusing. Please introduce a paragraph explaining it.

We have added an overview of the conceptual framework on p 12 lines 250-252. This text reads “This framework highlights the potential barriers and facilitating factors to acceptability at the individual/household, community, health system and intervention (i.e., MITS in this study) levels.” We did not analyze themes within the framework, but used it as a tool in our analysis and second round of FGDs.

ETHICS APPROVAL

Line 199: Include the reference number for the approvals (Protocol #17/09/1913, Reference 34-16, STUDY00003689), as you did in the Ethics Statement.

We have added the corresponding reference numbers for the approvals and exemption on p 13 lines 257-258, 260. The updated text reads “The Malawi National Health Sciences Research Committee (Protocol #17/09/1913) and the Oxford Tropical Research Ethics Committee (Reference 34-16) approved the study protocols. The University of Washington Institutional Review Board (STUDY00003689) exempted the study from review.”

RESULTS:

Line 229: Include which Malawian ethnic groups.

We have added specific reference to the Chewa group in southern Malawi, as they are most likely to seek care at QECH, on p 14 line 293 which now reads “

Maternal uncles are often regarded as the head of family lineage and frequently control decisions regarding children within the family in traditional, matrilineal systems found across some Malawian ethic groups, such as the Chewa of southern Malawi.”

Line 242: Quote. When the participant said that “(…) then there must be something that you know [about the child´s death]“ is she refereeing to something that the parents want to hide? As issues related with stigmatized diseases or mothers not taking care properly of their child? If yes, specify in line 237, where you are explaining blame.

A general sense that parents (not just mothers) might be blamed was conveyed by some participants if they consented without extended family support. No specific examples of issues related to stigmatized diseases or mothers not taking proper care of their children were related in this context. Only mothers in the first FGD round noted identification of stigmatized illnesses as a risk and they were concerned more about rumours within the community, rather than blame. 

Line 272: I do not understand the quote, when it reads “(…) with admission of this number of days”. Please, clarify.

We have clarified the meaning of this quote on p 16 lines 338-339. The quote now reads “they just write in files, they do not say what operation we have done, this is what we will do, with admission for this number of days [how long the child might be hospitalized following a medical procedure].”

Line 285: Very interesting. It would be good to describe in more detail which are the ordinary procedures explained by participants when someone dies at the hospital and who is responsible/does each one. For example: Who certifies the death? Who washes the body? Does it have a cost? Is the family allowed to wash the body themselves? How long since the person dies and the body is returned to the family?

Thank you, this information was not detailed by study participants except in relation to body washing which we have now clarified on p 17 line 356 which now reads “Parents and religious leaders highlighted existing rumors associated with routine hospital processes post-mortem, specifically holding the body in the morgue, washing of the body by mortuary staff, and organ harvesting to make medicines.”

Line 315: Which kind of resource constraints limit communication between HCW and parents? Are the authors referring to lack of staff, so they are overwhelmed and cannot attend the parents? About lack of a proper space at the hospital where HCW can talk with the parents privately and calmly? About no resources to improve personnel skills? Please, explain.

Specific resource constraints identified by HCWs included limited staff and time to attend to patient needs, which reduced their ability to effectively communicate with patients and families and foster trust. We have expanded on this in the manuscript draft on p 18 lines 381-383 which now reads “. However, HCWs highlighted that resource constraints, such as insufficient staffing and limited time to attend to patient needs, hamper their communication and ability to foster trust with parents.”

Line 322: Clarify in the quote who were “they”: “…they [parents? patients?] fear the doctor and nurses (…)

We have clarified on p 19 line 389 that this quote is referring to parents fearing doctors and nurses which now reads “they [parents] mostly fear the doctor and nurses…”

Line 329: Specify if all fathers in the FGD, almost all fathers, some or few.

We have specified this refers to most fathers on p 19 line 396 which now reads “However, this opinion was not shared by most fathers…”

Line 254, table 2: Benefits and risks associated with MITS.

- Which was the MITS in Malawi approach following formative research (column 3) for “Knowledge gained from MITS procedures could benefit the wider community” (column 1, row 5), if one?

MITS in Malawi is providing results to immediate family members and encouraged to disseminate to their broader community, if they are comfortable and if the findings might be protective for others. We have now specified this in Column 3, Row 5 of Table 2 on p 20 which reads “Information provided on initial COD 4-6 weeks post-death. Families encouraged to share findings within their social networks, if they desire. Plans for dissemination of study results (without identifiable information) broadly.”

- In the row “Disfigurement of the body” (column 1), the MITS in Malawi approach following formative research (column 3) is “Minimally Invasive Procedure”. Please explain. Did the protocol on how to perform MITS changed after this finding?

While this did not change the MITS protocol, it did inform how we explained the MITS procedures in the consent process, which is now noted on p 21 (Risks, row 1) as “Minimally invasive procedures will be described in consent process and families provided opportunity for view the body post-MITS.” In the informed consent form, we included specific details about the size of the puncture holes due to introduction of MITS needles and reassurances that we would not cut open, a stitch, or disrupt the body it in any other way. Additionally, we added to our procedures that families would be able to examine the body post-MITS, should they like to confirm no unexpected disfigurement.

- In the row “Potential identification of stigmatized illnesses that could cause challenges to families if revealed to community” (column 1), the MITS in Malawi approach following formative research was “strict confidentiality” (column 3). Please specify, both regarding the MITS´ results delivery strategy and regarding who are the only ones receiving the results. Parents? The ones that the parents have previously decided?

We have added additional information about how this finding shaped our procedures on p21 in the row “Potential identification of stigmatized illnesses that could cause challenges to families if revealed to community” (column 1). We have expanded on the approach text in column 3 (p 21) which now reads “Strict confidentiality. Results to be provided to family members who consented. In instances where vertical transmission of a stigmatized illness is identified, every effort will be made to first discuss with the mother privately.”

Line 356: * Transportation, coffin, and family support offered to all families approached for consent in MITS, not just those who participated. I would say instead: *Transportation, coffin, and family support to be offered to all families approached for consent in MITS, not just those who accept to participate.

Thank you, we have updated the text per your recommendation on p 21 lines 422-423 which reads “* Transportation, coffin, and family support to be offered to all families approached for consent in MITS, not just those who accept to participate.”

Line 402: Table 3: Reflections on appropriate forms of approach.

- I would suggest table 2 and 3 to maintain the same order for the columns (“Focus group noting the concept/suggestion” is placed in column 2 in the table 2 and in column 3 in table 3).

Thank you, we have updated table 3 per your suggestion on pp 23-24.

- I do not understand the text “Wait to sensitize until after hospital admission or risk parents leaving against medical advice” (column 2, row 3). Please, clarify.

We have clarified the text on p 23, column 3, row 2 to read “Conduct sensitization with parents/families after hospital admission and before death.” Participants from these groups did not perceive increased risk of families taking their child out of the hospital against medical advice with earlier sensitization, we have corrected the text to accurately reflect this as well.

- I would replace “[Avoid sensitization with] Risk parents leaving against medical advice” for “parents with perceived risk of them taking the child out of the hospital against medical advice” or similar. Is this what authors want to say?

We have updated the text in on p 24, column 3, row 3 to read “Avoid sensitization with parents/families in hospital before death to reduce the risk of parents/families taking the child out of the hospital against medical advice”

- “Complex and detailed MITS procedure information will not translate well in Chichewa and might worry participants” (Approach factor. Consent, column 2): This is not a suggestion.

Thank you for pointing this out. We have removed this text in Table 3 on p 24.

DISCUSSION:

Line 479. I would replace “Our study adds data…” for “Our study adds knowledge…”.

We have updated the text per your suggestion on p 28 line 543 which now reads “Our study adds knowledge about the centrality of social relationships to pediatric MITS acceptability, including the roles of trust, power, and communication within relationships.” 

REFERENCES:

Line 499: Reference 2: Add when data were retreived: 2. UNICEF. Under-five mortality: Child mortality data 2019 [Available from: https://data.unicef.org/topic/child-survival/under-five-mortality/. Accessed on XXX]

We have added the access date to this reference on p 29 line 565. 

Line 579: Delete reference 31.

We have removed the QSR International reference for NVivo.

---

## [Decision Letter · Decision Letter 1]

11 Jan 2021

PONE-D-20-25933R1

Hypothetical acceptability of hospital-based post-mortem pediatric minimally invasive tissue sampling in Malawi: The role of complex social relationships

PLOS ONE

Dear Dr. Lawrence,

Thank you for submitting your manuscript to PLOS ONE. After careful consideration, we feel that it has merit but does not fully meet PLOS ONE’s publication criteria as it currently stands. Therefore, we invite you to submit a revised version of the manuscript that addresses the points raised during the review process.

We look forward to receiving your revised manuscript.

Kind regards,

Ritesh G. Menezes, M.B.B.S., M.D., Diplomate N.B.

Academic Editor

PLOS ONE

Reviewers' comments:

Reviewer's Responses to Questions

**Comments to the Author**

1. If the authors have adequately addressed your comments raised in a previous round of review and you feel that this manuscript is now acceptable for publication, you may indicate that here to bypass the “Comments to the Author” section, enter your conflict of interest statement in the “Confidential to Editor” section, and submit your "Accept" recommendation.

Reviewer #1: All comments have been addressed

Reviewer #2: All comments have been addressed

Reviewer #3: All comments have been addressed

2. Is the manuscript technically sound, and do the data support the conclusions?

Reviewer #1: Yes

Reviewer #2: Yes

Reviewer #3: Yes

3. Has the statistical analysis been performed appropriately and rigorously? 

Reviewer #1: Yes

Reviewer #2: Yes

Reviewer #3: N/A

4. Have the authors made all data underlying the findings in their manuscript fully available?

Reviewer #1: Yes

Reviewer #2: Yes

Reviewer #3: Yes

5. Is the manuscript presented in an intelligible fashion and written in standard English?

Reviewer #1: Yes

Reviewer #2: Yes

Reviewer #3: Yes

6. Review Comments to the Author

Reviewer #1: The manuscript addresses a critical issue related to the MITS conduct in Malawi. The formative research findings must be used for shaping the conduct of the MITS and obtaining consent for MITS. The authors should comment on use of the findings in the discussion section.

Reviewer #2: Well done on revisions. Two minor suggestions:

Line 117 is 'expectances' a typo? Unsure of the meaning of that word.

Line 443: incomplete or interrupted sentence immediately following Table 3

Reviewer #3: I have reviewed the manuscript and all suggestions have been addressed. There is a mistake in line 269: It reads “ethic groups” instead of “ethnic groups”. I am satisfied with the current version of the manuscript.

7. PLOS authors have the option to publish the peer review history of their article (what does this mean?). If published, this will include your full peer review and any attached files.

Reviewer #1: **Yes: **Manoja Kumar Das

Reviewer #2: No

Reviewer #3: No

---

## [Author Response · Author response to Decision Letter 1]

15 Jan 2021

Reviewer 1

- The manuscript addresses a critical issue related to the MITS conduct in Malawi. The formative research findings must be used for shaping the conduct of the MITS and obtaining consent for MITS. The authors should comment on use of the findings in the discussion section.

Thank you for this comment, in the discussion section we note the following on lines 513-516: “…this formative research provided a foundation for implementation of a MITS study to elucidate CoD among hospitalized young Malawian children. Important lessons learned informed development of consent and study procedures.” Details of our specific approaches are outlined in Table 2 (pp 19-20) in the 3rd column. 

Reviewer 2

- Well done on revisions. Two minor suggestions:

Line 117 is 'expectances' a typo? Unsure of the meaning of that word.

Thank you for this feedback, we have updated the word “expectances” to read “expectations” in line 117. The full text now reads “Furthermore, consent and reported satisfaction reflect power relations and expectations more than acceptance.”

- Line 443: incomplete or interrupted sentence immediately following Table 3

Thank you, we have deleted this text from line 443.

Reviewer 3

- I have reviewed the manuscript and all suggestions have been addressed. There is a mistake in line 269: It reads “ethic groups” instead of “ethnic groups”. I am satisfied with the current version of the manuscript.

Thank you, we have updated the text appropriately to read “ethnic groups” in line 269.

---

## [Editor Report · Decision Letter 2]

19 Jan 2021

Hypothetical acceptability of hospital-based post-mortem pediatric minimally invasive tissue sampling in Malawi: The role of complex social relationships

PONE-D-20-25933R2

Dear Dr. Lawrence,

We’re pleased to inform you that your manuscript has been judged scientifically suitable for publication and will be formally accepted for publication once it meets all outstanding technical requirements.

Kind regards,

Ritesh G. Menezes, M.B.B.S., M.D., Diplomate N.B.

Academic Editor

PLOS ONE

---

## [Editor Report · Acceptance letter]

22 Jan 2021

PONE-D-20-25933R2 

Hypothetical acceptability of hospital-based post-mortem pediatric minimally invasive tissue sampling in Malawi: The role of complex social relationships 

Dear Dr. Lawrence:

I'm pleased to inform you that your manuscript has been deemed suitable for publication in PLOS ONE. Congratulations! Your manuscript is now with our production department. 

Kind regards, 

on behalf of

Prof. Dr. Ritesh G. Menezes 

Academic Editor

PLOS ONE